

# The regional climate model REMO (v2015) coupled with the 1-D freshwater model FLake (v1): Fenno-Scandinavian climate and lakes

Joni-Pekka Pietikäinen[1], Tiina Markkanen[1], Kevin Sieck[2], Daniela Jacob[2], Johanna Korhonen[3], Petri Räisänen[1], Yao Gao[1], Jaakko Ahola[1], Hannele Korhonen[1], Ari Laaksonen[1], and Jussi Kaurola[1]

[1]Finnish Meteorological Institute, P.O. Box 503, 00101, Helsinki, Finland
[2]Climate Service Center Germany, Chilehaus - Eingang B, Fischertwiete 1, 20095, Hamburg, Germany
[3]Finnish Environment Institute SYKE, P.O. Box 140, 00251, Helsinki, Finland

*Correspondence to:* Joni-Pekka Pietikäinen (Joni-Pekka.Pietikainen@fmi.fi)

**Abstract.** The regional climate model REMO was coupled with the FLake lake model to include an interactive treatment of lakes. Using this new version, the Fenno-Scandinavian climate and lake characteristics were studied in a set of 35-year hindcast simulations. Additionally, sensitivity tests related to the parameterization of snow albedo were conducted. Our results show that overall the new model version improves the representation of the Fenno-Scandinavian climate in terms of 2-m temperature

and precipitation, but the downside is that an existing wintertime cold bias in the model is enhanced. The lake surface water temperature, ice depth and ice season length were analyzed in detail for ten Finnish, four Swedish, two Russian and one Estonian lakes. The results show that the model can reproduce these characteristic with reasonably high accuracy. The cold bias during winter causes e.g. overestimation of ice layer thickness at several of the studied lakes, but overall the values from the model are realistic and represent well the lake physics in a long-term simulation. We also analyzed the snow depth on ice

from ten Finnish lakes and vertical temperature profiles from five Finnish lakes and the model results are in realistic.

## 1  Introduction

The interactions between the atmosphere and the underlying surface are among the most important factors in climate and numerical weather prediction (NWP) modelling (Mironov, 2008; Samuelsson et al., 2010). Land and sea surfaces are dominant globally, but regionally, lake surfaces can play a significant role. Locations like northern Europe, Asia and North America have

rather high lake area densities, and the interactions between the atmosphere and lake surface can be a major driver in regional and local climate. Thus, in order to make reliable climate simulations, models should include lake modules that have reasonable complexity to simulate interactions between the atmosphere and lakes (Kirillin et al., 2012).

In regional climate models (RCM), lakes have been historically taken into account by setting the related variables (e.g. surface temperature and ice conditions) to follow external data, which often has been derived from the same data source

as the lateral boundary data for the atmospheric variables. This approach can be applied in regions with a low fractional area of lakes whereas regions with a large fractional area of lakes suffer from the limited interactions between lakes and the





atmosphere. Deficiencies or missing representation of lake processes can cause artificial features in climate model results. For example, Kotlarski et al. (2014) showed that over Fenno-Scandinavia many EURO-CORDEX (Coordinated Regional Climate Downscaling Experiment for European domain) RCMs have artificial heat sources during winter. These can be directly linked to the large fractional area of lakes, whose prescribed surface temperatures violate surface-atmosphere interactions. Some of

the models still use a simple approach where the lake temperatures and ice conditions are linked to the closest sea point. In reality, Fenno-Scandinavian lakes usually freeze before the nearest sea area, because there the lakes are more shallow and consist of freshwater (Kirillin et al., 2012). This means that with the nearest sea point approach, lakes continue to emit heat and moisture to the surrounding atmosphere for far too long during early winter, thus causing artificial heat and moisture fluxes to the atmosphere.

To overcome the deficiencies in lake representation, lake models have been implemented in some RCMs (e.g. Martynov et al., 2010; Samuelsson et al., 2010; Gula and Peltier, 2012; Mironov et al., 2012; Bennington et al., 2014) and NWP models (e.g. Mironov et al., 2010; Yang et al., 2013). In most cases these models resolve the vertical profiles of lake variables in 1-D without any direct horizontal influence. This approach is widely used, because it is computationally efficient and has been shown to reproduce realistic lake and climate related variables (Kourzeneva et al., 2012; Stepanenko et al., 2013; Mallard et al.,

2014). There are also 2-D and 3-D lake models (e.g. León et al., 2007), but these models are usually much more computationally demanding than 1-D models. Furthermore, 3-D lake models require a higher resolution computational grid (∼2 km) and lake specific inflow and outflow information (Martynov et al., 2010). In principle, a high resolution 3-D lake model could be coupled with a RCM of lower resolution by various down-scaling approaches, but this would further increase the computational costs.

One widely used 1-D freshwater lake model is FLake (Mironov, 2008), which has been coupled into several climate and

weather models (Stepanenko et al., 2013). For example, Mironov et al. (2010) introduced FLake into the numerical weather prediction model COSMO and showed that the coupling improved the prediction of lake surface temperatures, the freeze-up of lakes and the ice break-up across Europe. The COSMO-FLake system eliminated the significant overestimation of the lake surface temperatures during winter. Samuelsson et al. (2010) showed that the implementation of Flake to the Rossby Centre regional climate model RCA reduces known biases by increasing the 2m temperature over Europe by up to 1 °C and the

precipitation by 20-40% when compared to a simulation without lakes. Martynov et al. (2012) simulated North American climate with the Canadian Regional Climate Model (CRCM) using two different lake models: FLake and Hostetler (Hostetler and Bartlein, 1990). The authors showed that FLake performed very well over their domain giving better results than the Hostetler model while the large and deep lakes caused some problems for both models. Finally, Mallard et al. (2014) showed that in the Weather Research and Forecasting (WRF) model, FLake significantly improved the timing and extent of ice coverage

and reduced errors in lake temperatures.

In this study, FLake has been interactively coupled with the regional climate model REMO. With the new model version called REMO-FLake, we have simulated over 35 years of the Fenno-Scandinavian climate and compared the results against the default model version and measurement data. Moreover, the lake related variables (e.g. surface temperature, ice thickness and snow depth on ice) have been compared to observations for ten Finnish, four Swedish, two Russian and one Estonian lakes.

Additionally, some analysis has been made for vertical temperature profiles in five Finnish lakes.





The article is structured as follows: first, REMO, FLake and the implementation structure are described in Section 2; in Sections 3 and 4 a detailed analysis of the results is given; finally, in Section 5, the main conclusions are discussed.

## 2   Methods

### 2.1   REMO

In this work, the hydrostatic version of the REgional MOdel REMO (version REMO2015) has been used (Jacob and Podzun, 1997; Jacob, 2001). REMO is a three-dimensional atmosphere model developed at the Max Planck Institute for Meteorology in Hamburg, Germany, and currently maintained at the Climate Service Center Germany (GERICS) in Hamburg. The model is based on the Europa Model, the former numerical weather prediction model of the German Weather Service. The prognostic variables in REMO are horizontal wind components, surface pressure, air temperature, specific humidity, cloud liquid water

and ice. The physical packages originate from the global circulation model ECHAM4 (Roeckner et al., 1996), although many updates have been introduced (e.g. Hagemann (2002); Semmler et al. (2004); Pfeifer (2006); Kotlarski (2007); Rechid (2009); Teichmann (2010); Pietikäinen et al. (2012); Preuschmann (2012); Wilhelm et al. (2014)).

REMO uses a leap-frog scheme with time filtering by Asselin (1972) for the temporal discretization. To allow for longer time steps a semi-implicit correction is used. REMO's vertical atmospheric levels are represented in a hybrid sigma-pressure

coordinate system, which follows the surface orography in the lower levels and is independent of it at higher atmospheric model levels. Horizontally, REMO has a spherical Arakawa-C grid (Arakawa and Lamb, 1977), where all prognostic variables, except winds, are calculated at the center of a grid box. The wind components are calculated at the edges of the grid boxes.

At the lateral boundaries of the model domain, REMO uses the relaxation scheme developed by Davies (1976). In this scheme, the prognostic variables of REMO are adjusted towards large-scale forcing at the 8 outermost grid boxes. The outside

forcing decreases exponentially in this zone towards the inner model domain. At the surface, the sea surface temperature (SST) and sea ice distribution are prescribed by the lateral boundary forcing data-set. At land points, a fractional land surface scheme is used and details can be found, for example, in Kotlarski (2007) and Rechid (2009). Shortly, the scheme uses a tile approach with 3 tiles: land, water and ice.

Lakes are included in REMO through the default land cover map: Global Land Cover Characteristics Database (GLCCD;

Loveland et al., 2000; US Geological Survey, 2001). However, the standard land surface scheme does not have a tile for lakes and thus does not allow for their explicit treatment. Instead, the model uses the nearest sea-point approach for all non-sea water fractions to determine the water temperature and ice conditions. As indicated above, the nearest sea-point approach can lead to a significant distortion of the simulated climate, which suggests that REMO would benefit from a coupling with a physical/process based lake model.



## 2.2 FLake lake model

FLake is a thermodynamic freshwater lake model, which predicts the mixing conditions and vertical temperature structure of lakes on time scales from a few hours to several years (Mironov, 2008; Mironov et al., 2010). FLake's water module calculates the heat and kinetic energy budget for the upper mixed-layer and the basin bottom. The model can be used for various basin

depths and even though it has been intended for use in NWP and climate models, it can be used as a standalone model as well. FLake uses a bulk-approach and is based on a self-similarity (assumed-shape) representation of the temperature profile in the mixed-layer and thermocline. In addition, it calculates the mixed-layer depth as well as the temperature and thickness of both ice and snow on ice. Optionally, FLake can calculate the flux between the bottom sediment and the lake bottom. In this case, thickness and temperature of the thermally active upper sediment layer are calculated. A more detailed description of FLake

can be found in Mironov et al. (2010), where the authors give a detailed list of FLake parameters and more information about on the numerical core.

## 2.3 Implementation of the FLake model

REMO's surface fractions are based on the Global Land Cover Characteristics (GLCC) database (US Geological Survey, 2001). This database has a nominal 1-km spatial resolution and includes lakes as one variable. REMO's surface pre-processor, which

creates the surface related parameters including the fractions of different tiles (previously only land, (sea) water and ice), was modified in this work to include also the lake fraction as an output variable. The lake depth data were taken from the detailed dataset developed by Choulga et al. (2014), which gives the mean depths of lakes in a global grid (version v3). Using a dataset with realistic lake depths is important, as shown by Samuelsson et al. (2010). Additionally, the dataset by Choulga et al. (2014) includes lake fractions, but to be consistent with the rest of the model, the GLCC lake fractions were used. In practice, this

means that whenever there is a lake in the GLCC dataset, the lake depth is taken from the Choulga et al. (2014) dataset. For points where Choulga et al. (2014) has no data, a default value is used. The default value can be set beforehand within the surface pre-processor and for example, was set to 7 m in this work. The value is based on data by Choulga et al. (2014) for the regions 126, 130 and 132 that fall within the present calculation domain used in this work.

    FLake is directly implemented to REMO so that in every time step all the lake-related parameters are calculated online. The

lake tile is assumed to be either completely open (ice fraction 0) or completely frozen (ice fraction 1). This differs from the original water area approach, which is separated into water and ice tiles. This is due to the 1-D approach of FLake, which does not distinguish between different lakes within one gridbox. In practice, this means that the lake tile has three possible states: open water, ice-covered without snow, or ice-covered with snow on the ice. All the tile-specific variables (e.g. specific humidity, surface temperature, surface fluxes and albedo) are now calculated for the lake tile by taking into account the tile's

state.

    The FLake model also includes a surface-flux parameterization (SfcFlx). However, as REMO can produce all the necessary surface-related variables and fluxes, SfcFlx has not been implemented. Below the land surface, REMO uses a diffusion equation solver in a 5-layer model with zero heat flux at the bottom (10m). For the lake tile, this has not been used; instead, FLake's





own lake bottom sediment module has been implemented to REMO. It calculates the heat flux at the water-bottom sediment interface by using the depth of the upper layer of bottom sediments that is penetrated by the thermal wave, and the temperature at that depth. Moreover, as FLake is not suitable for deep lakes (Mironov et al., 2010), a "false bottom" approach has been used. In this approach, the maximum lake depth has been set to 50 m and the bottom sediment module has been switched off

for lakes deeper than this. This has a negligible impact on the results, because typically there are no significant temperatures changes in such depths for fresh-water lakes (Samuelsson et al., 2010).

Even though the snow module of FLake has not been thoroughly tested, there is some evidence that especially in climate simulations the snow module gives realistic results (D. Mironov, personal communications, 2015). We have also updated the coefficients used to calculate snow density and heat conductivity as suggested by Mironov et al. (2012). These changes are

needed as the original approach yielded too low snow density and too high snow temperature conductivity. Moreover, we have implemented an alternative approach for snow heat conductivity, with an effective snow heat conductivity $k_{\text{seff}}$ as proposed by Semmler et al. (2012):

$$k_{\text{seff}} = k_{\text{s}} + (k_{\text{i}} - k_{\text{s}}) \times e^{-hs \times c}, \qquad (1)$$

where $k_{\text{s}}$ is the heat conductivity of snow (0.14 W/(mK)), $k_{\text{i}}$ is the heat conductivity of lake ice (2.29 W/(mK)) and $c$ is

an empirical constant (5 m$^{-1}$). In practice, this approach increases the heat conductivity for thin snow, because it is highly probable that there is also bare ice in the vicinity of thin snow due to snow redistribution by wind drift; thus, the $k_{\text{seff}}$ is actually a mixture of snow and ice conductivity. Due to stability issues, we start to accumulate snow over ice only when the ice layer is at least 3 cm thick. In addition, the ice albedo limits from Semmler et al. (2012) have been implemented (similar changes were also made by Yang et al. (2013)). The default albedos in FLake is from 0.1 tp 0.6 for ice and snow, respectively, while the ice

albedo range applied in Semmler et al. (2012) is from 0.3 to 0.5. Moreover, we have changed the snow albedo limits and these changes will be shown in details in the next section.

## 2.4  The snow albedo

Three treatments of snow albedo $\alpha_{sn}$ are considered in this work: a temperature-dependent scheme, a snow albedo scheme originating from the Biosphere-Atmosphere Transfer Scheme (BATS; Dickinson et al., 1993), and their combination. By default,

REMO employs a temperature-dependent snow albedo scheme. The shortwave radiation module for REMO separates visible and near-infrared spectral regions (VIS and NIR, respectively) for surface albedo, but in the default treatment, a broadband approach (the same albedo for the VIS and NIR regions) is used for snow albedo. The snow albedo over land is temperature dependent and gets its maximum value $\alpha_{sn_{max}} = 0.8$ when the snow temperature is below $-10$ °C. At higher temperatures $\alpha_{sn}$ decreases linearly until it reaches the snow albedo minimum $\alpha_{sn} = 0.4$ at the freezing point. Moreover, the forest fraction

of the grid cell influences linearly the albedo maximum and minimum values so that they reach values of $\alpha_{sn_{max}} = 0.4$ and $\alpha_{sn_{min}} = 0.3$ at a forest fraction of unity (details in Kotlarski (2007)). However, these values of forested $\alpha_{sn_{max}}$ and $\alpha_{sn_{min}}$ are slightly higher than those shown in the literature (e.g. Roesch et al., 2001, and references therein), and therefore, in the





present work we have reduced both $\alpha_{sn_{max}}$ and $\alpha_{sn_{min}}$ to 0.25 for completely forest-covered gridboxes, following Gao et al. (2014).

The radiation module of REMO separates visible and near-infrared light regions (VIS and NIR, respectively) for the albedo. However, the snow albedo does not change between these regions which means that the albedo is a so-called broadband

albedo. In REMO, the snow albedo over land is temperature dependent and gets its maximum value $\alpha_{sn_{max}} = 0.8$ when the snow temperature is below 10 °C. At higher temperatures the $\alpha_{sn}$ linearly decreases until it reaches the snow albedo minimum $\alpha_{sn} = 0.4$ at the freezing point. Moreover, the forest fraction of the grid cell influences linearly the albedo minimum and maximum values so that they reach values $\alpha_{sn_{max}} = 0.4$ and $\alpha_{sn_{min}} = 0.3$ at a forest fraction of unity (details in Kotlarski (2007)). However, the values of forested $\alpha_{sn_{max}}$ and $\alpha_{sn_{min}}$ are slightly higher than those shown in literature (e.g. Roesch

et al., 2001, and references therein), and therefore, in the present work we have reduced both $\alpha_{sn_{max}}$ and $\alpha_{sn_{min}}$ to 0.25 for gridboxes covered with forest only, following Gao et al. (2014).

The temperature-dependent $\alpha_{sn}$ takes into account that the snow albedo is lower when snow is wet, i.e. is near the freezing point. However, this approach does not take into account the aging of snow, soot loading, grain size or the influence of solar zenith angle. As these play a role in $\alpha_{sn}$, in this work, we have implemented the snow albedo scheme from the Biosphere-

Atmosphere Transfer Scheme (BATS; Dickinson et al., 1993). In this scheme, $\alpha_{sn}$ takes into account the aging of snow and solar zenith angle, and is calculated separately for VIS and NIR. The aging of snow is based on three terms: the first represents the grain growth effect due to vapor diffusion and it reflects the effect of grain growth due to vapor diffusion, the second represents the additional effect near and at freezing of melt water, and finally the third one introduces the influence of soot and dirt through a global time-independent constant (details in Dickinson et al. (1993)). The VIS $\alpha_{sn}$ is allowed to get values

between 0.9 and 0.5 and the NIR $\alpha_{sn}$ between 0.65 and 0.25, where the highest values are employed for new snow. The forest fraction decreases these values as with the temperature-dependent broadband approach so that a fully forested gridbox has a VIS snow albedo of 0.25 and a NIR albedo of 0.2.

The temperature-dependent scheme aims to describe the reversible changes in the crystal structure of snow when the temperatures approach the melting point. On the other hand, the BATS scheme describes the irreversible crusting of a snow layer

and the accumulation of aerosols and other impurities in the snow through the aging factor. As these both are important in reality, we modified the source code so that one can choose either the original scheme, the new BATS scheme, or a combined approach similar to that employed in the JSBACH model (Raddatz et al., 2007; Brovkin et al., 2013; Reick et al., 2013). In the JSBACH approach, the schemes are weighted and in this work we have used equal weights.

The albedo reduction due to forests only occurs over land areas. Therefore, snow albedo (and for the BATS scheme, snow

aging) is calculated separately for land and lake tiles as one gridbox can have snow on both tiles. The gridpoint snow albedo is then calculated as an area-weighted mean value of the tile albedos.

## 2.5 Simulations

We have made simulations for the Northern Europe for the period of August 1979 – March 2015. The simulations employed a warm-start method, which means that at the start of the simulation soil temperature, soil moisture and lake variables were



obtained from existing simulation data. In the warm-start method, the model was run for the whole time period with default initial values (for REMO and REMO-FLake separately) and then using the model's final state as the initial state for the actual and analyzed simulations. More details about the method for the same domain can be found in Gao et al. (2014) and Gao et al. (2015).

The horizontal resolution of the simulations was 18 km×18 km (0.167°) with 27 vertical layers (the model top reaching 25 km altitude) and 90 s time step. The lateral meteorological boundary forcing was obtained from ERA-Interim data with a 6-hourly update frequency (Dee et al., 2011). The simulation domain can be seen in Fig. 1, where the used lake fraction, mean depth, surface elevation, and locations and names of the analyzed Finnish lakes are shown. We have conducted a total of 5 simulations: 1 without FLake and 4 with FLake (Table 1). The simulation without FLake (REMO-ST) was conducted using

the default configuration of the model. The simulations with FLake included a baseline run (REMO-FL) with the temperature-dependent snow albedo scheme, a run with the BATS snow albedo scheme (REMO-FLB), and a run with the combined temperature-BATS albedo scheme (REMO-FLTB). The fourth simulation (REMO-FLOS) was otherwise similar to REMO-FL but used FLake's original snow heat conductivity approach instead of the Semmler et al. (2012) approach. For all temperature-dependent snow albedo simulations (REMO's default configuration) the broadband albedo has been used, but for all simulations

including the BATS scheme, the VIS and NIR separation approach has been used (details in Section 2.4).

## 2.6  Lake data

The modelled lake variables are evaluated against three different measurement data. Firstly, the main data source covering all analyzed lakes in Finland is the measurement data from the Finnish Environment Institute (SYKE). Measurement data of SYKE includes lake surface water temperatures (LWT), ice cover periods and ice thickness with snow on ice for all studied lakes.

Vertical water temperature profiles are available for five lakes. Secondly, the International Lake Environmental Committee (ILEC) data is used for lake surface temperatures and ice cover period outside Finland (ILEC, 1988-1993). Lastly, for some lakes in Sweden, Russia and Estonia the ice cover period is taken from the National Snow and Ice Data Center's (NSIDC) Global Lake and River Ice Phenology data (Benson and Magnuson, 2000, updated 2012).

   The Great Saimaa Lake is the largest lake by area in Finland (see Table 2 and Fig. 1). However, its areal definition is not

unambiguous, because it is a combination of several open lake areas connected by straits. In our study, we have divided the Great Saimaa Lake into two parts. Lake Haukivesi is a lake in between Varkaus and Savonlinna cities and represents the north-western part of the Great Saimaa Lake. Another focus area in our study is the southern part, where the open lake area is actually called Saimaa, i.e. in our study, the name Lake Saimaa points to the southern area of the Great Saimaa Lake.

   Table 2 shows the mean lake depths and the means calculated from the data by Choulga et al. (2014). It also shows the

longitudinal and latitudinal boundaries that are used to define the lake area in the analysis of the model results. As can be seen, the average depths are reasonably similar between the literature and model data, except at Lappajärvi, where the dataset by Choulga et al. (2014) gives a 16 m depth for the whole lake while in reality it is closer to 7 m, and at Kallavesi, Haukivesi and Näsijärvi, where the FLake database does not have information about the depth and the default value of 7 m is used, at Mälaren, where the overall shape of the lake is not well represented in the database and causes underestimation of the mean

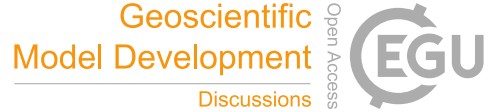



depth, and at Vättern, where the difference is slightly over 10 m. There is also over 10 m difference at northern part of lake Ladoga, but there the literature based average depth for the whole lake has been also used for the northern part and this causes error.

Table S1 (supplementary) introduces the measurement locations from the SYKE database (the locations can be seen from

Fig. 1). It defines the actual locations (lake name is used if the measurement is done at the open lake or if it represents the whole lake area) and shows how many measurements sites are used in the actual analysis. If an analyzed variable has more than one source for some time period, a mean value has been used. Different sources give more information about the spatial differences of the analyzed area, which reduces the error when comparing point measurements against larger model grids. The LWTs are measured at 8 o'clock in the morning (local time) and all modelled LWTs are also taken from this same time of the

day. Ice thickness and the snow depth on ice have been measured at different times of the day and thus the modelled daily mean value is used. The SYKE measurements for LWT, ice depth and snow on ice are done close to the lake shore, which causes some error when compared to lake's average modelled values. This will be discussed in more details in the following sections.

## 3    Climate impacts

We have compared the modelled 2-m air temperature and precipitation fields against the E-OBS version 16 gridded dataset

(Haylock et al., 2008). For this comparison, the modelled values were remapped to the E-OBS grid, which is slightly coarser (0.25°) than the REMO grid. The accuracy of the E-OBS data depends on the density of the underlying measurement network and naturally this introduces some error (e.g. Kotlarski et al., 2014, and references therein). We do not explicitly consider the observational uncertainties like undercatch of precipitation, but they should be kept in mind (Prein and Gobiet, 2017). Moreover, the relaxation zone, i.e. the area where the lateral forcing directly influences the results, has been removed from the

analysis.

### 3.1    Influence of the lake model FLake

The comparison of the model results against the E-OBS 2-m air temperature and surface precipitation for the standard REMO run (REMO-ST) and for the REMO-FLake (REMO-FL; details in Tab. 1) is shown in Fig. 2. As the E-OBS data is based on measurements over land and the values given over lakes are an interpolation of these measurements, a direct comparison

against the model's results causes artificial biases over gridboxes with a high lake fraction. To reduce these biases, we have excluded all grid points where the lake fraction is equal or larger than 0.5. While we have no direct measurements of 2-m temperatures over lakes, we will analyze in Sections 4.1 and 4.2 the LWTs, which are generally close to the 2-m temperatures over lakes. Here, we only show the biases of the model versions against the E-OBS data, while the absolute E-OBS values for $T_{2m}$ and precipitation are shown in the supplementary Fig. S1.

During the winter season, REMO-ST shows an overall temperature bias of -0.9 K for the domain while REMO-FL has a -2.4 K bias, as can be seen from Fig. 2. The spatial distribution of $T_{2m}$ in REMO-ST shows that the model has a cold bias in the east and warm biases in northern Finland and throughout Sweden. The reason for the cold bias is still unknown, but the warm



biases originate from the treatment of lakes. While in reality the lakes should be frozen, in the model they are open due to the nearest sea point treatment. This causes artificial heating and shows as warm biases in the spatial maps. With REMO-FLake, the warm bias is gone as the model now interactively calculates the LWTs and ice periods. This, on the other hand, leads to a very strong cold bias throughout the domain. Our analysis (not shown here) indicates that the bias only occurs when there is

snow on the ground, which strongly suggests that the cause of the bias is linked to the surface treatment of the model rather than, for example, missing low-level clouds (without totally ruling out this possibility).

The spring season biases follow qualitatively the winter results. REMO-ST has a smaller cold bias due to warmer temperatures over lakes (-0.7 K). REMO-FL increases the cold bias to -1.4 K as the lakes stay frozen longer than in REMO-ST and do not artificially pump heat to the atmosphere. The cold bias is weaker than in winter as the fraction of snow covered surfaces

decreases. During summer, REMO-ST has a small cold bias over the domain and REMO-FL a slight warm bias. Overall, the summer temperatures are captured very well in both model versions. Autumn temperatures are warm biased in REMO-ST, but much better captured by REMO-FL. The REMO-ST biases primary occur near lakes, which indicates that the original approach for lakes starts to heat the atmosphere unrealistically while in reality lakes cool faster.

Figure 2 also shows the precipitation biases of REMO-ST and REMO-FL against E-OBS observational data. REMO-ST has

a winter bias of 22.7% over the whole domain while REMO-FL has a 13.5% bias. The difference between the model versions is coming from the areas with high lake density. Although the biggest lake areas have been masked out, due to transport processes the excess in moisture that triggers precipitation will show up also near the masked areas. This is visible especially in northern Finland, where Lake Inari is located. It is clear that the use of FLake removes the artificial heat and moisture sources and thus decreases the wet bias. The precipitation biases during spring are also smaller with FLake (REMO-ST 54.5% and REMO-FL

47.3%) and the largest biases occur in northern Finland and the surrounding areas. The difference to winter is that a wet bias occurs throughout the domain indicating that REMO captures better the snow precipitation rates than warm precipitation rates, i.e. rain.

The use of FLake increases significantly the precipitation bias in summer. REMO-ST has a 28.4% bias, but REMO-FL yields a 46.6% bias with largest excess in precipitation in central-eastern Finland. Some overestimation can be also seen

over Sweden. The increased wet bias seems to be connected to the fraction of lakes (Fig. 1) and is caused by the increased convective precipitation (analysis not shown here). With the current horizontal resolution (18km×18km), the model uses a convective parameterization to calculate the convection processes and it is based on the mass-flux scheme from Tiedtke (1989) with modifications by Nordeng (1994). With REMO-FL, the lake surfaces are warmer and thus there is more evaporation and humidity, which tends to trigger the convection. Although not shown here, we analyzed the lake surface temperatures also from

the lateral boundary data, which are the ones standard REMO uses. Our analysis showed that the LWTs in REMO-FL are 3-7 °C higher on average during summer than those in REMO-ST. This explains why there is more heat available for triggering the convection over lakes in REMO-FL. Nevertheless, we have not investigated in detail why there is excess in convective precipitation and only suggest possible reasons. The convection scheme works during a single time step, i.e. convective clouds form and precipitate during one time step, which causes precipitation to be too localized over lakes and there is not sufficient

transport of moisture. Another factor might be that the cloud cover is too small in the model, because convective clouds are not





radiatively active (i.e. the radiation scheme does not "see" convective clouds directly, but the model accounts for their transport of moisture in the large-scale cloud scheme). Naturally, the LWTs in REMO-FL could be too high, but this does not seem to be the reason, as will be seen in the later analysis.

The precipitation bias for REMO-ST during autumn (22.6%) is higher than for REMO-FL (17.8%). Like in winter, the wet bias in REMO-ST originates from lake areas: in reality, lakes cool faster than with the nearest sea point approach and this leads to excess in heat and moisture near lakes in the nearest sea point approach. When FLake is used, temperature and moisture are more realistic leading to smaller biases during autumn.

## 3.2 Snow analysis

We have compared the model's albedo to the CLARA-A2 dataset (Karlsson et al., 2017), which contains a global dataset of surface radiation products based on the measurements of the Advanced Very High Resolution Radiometer (AVHRR) on-board the polar orbiting National Oceanic and Atmospheric Administration (NOAA) and MetOP satellites (Schulz et al., 2009; Karlsson et al., 2012; Riihelä et al., 2013). Since the satellite dataset does not cover mid-winter months due to too low solar elevations, we have compared the monthly mean albedos for March and April. During these months, our domain still has snow and additionally, we can see the impacts of the snow melting period. Moreover, as the purpose of the comparison is to show how much the different snow albedo methods influence the results, we limit the comparison to land and lake surfaces.

Figure 3 shows the multi-year monthly measured albedos and the model bias (model-measurements) for the time period of 1982 to 2014. We compare three model versions, which are the default REMO-FLake (REMO-FL), REMO-FLake with the BATS albedo scheme (REMO-FLB) and REMO-FLake with the combined BATS and temperature albedo scheme (REMO-FLTB). Based on the albedo values, there is quite a lot of snow in our domain during March, whereas during April, the southern part of the domain is snow-free. The different model versions show some differences in March; for example, over the central-eastern part of the domain, REMO-FLB gives higher albedos than the simulations using the original (REMO-FL) or the combined albedo parameterization (REMO-FLTB). Overall over land, REMO-FL tends to underestimate the albedo when there is snow, but it is quite close to the measurements otherwise. The albedo over lakes in March is fairly well captured, although slightly overestimated with REMO-FLTB and especially in REMO-FLB. In March, the lake albedos seem to be overestimated in all model versions, although with REMO-FL only slightly. The overestimation is due to the cold bias in the model, which delays the melting of snow and ice. The same features can be seen as in April: the albedo in regions with snow is slightly underestimated. We also analyzed how much the changes made to the snow albedo in forested regions impact the results (see Sec. 2.4), but the difference is insignificant and cannot explain the overestimation (details not shown here).

The impact of the different snow albedo parameterizations on simulated 2-m temperature and precipitation can be seen by comparing REMO-FL in Fig. 2 with REMO-FLB and REMO-FLTB in Fig. S2. Overall, the impact is small. In the experiment with the temperature-based snow albedo scheme (REMO-FL), snow melts somewhat faster than in the other two experiments, but the domain-mean differences in seasonal-mean $T_{2m}$ bias are quite small, within $\approx 0.2$ K. A partial explanation for this is that as the model domain is quite northerly, the intensity of solar radiation is relatively low especially during winter. This

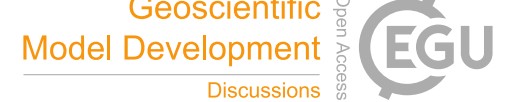

suggests that the cause of the bias is more linked to the thermodynamics of the snowy surface or boundary layer treatment in the model, although the possibility of missing low-level clouds cannot be excluded.

We have also made some sensitivity experiments to see the impact of the snow heat conductivity. The changes by Semmler et al. (2004), applied only over lakes, give better results than the original approach in REMO-FLOS (see Supplementary Fig.
S2). The mean over the domain during winter does not differ significantly (REMO-FL -2.4 K vs. REMO-FLOS -2.7 K), but from supplementary Fig. S2 we can see that the spatial pattern in the bias changes quite a lot. This also indicates that thermodynamics of the snow layer can play a big role in the cold-bias problem. The differences in precipitation are also small throughout the whole year. Overall, REMO-FLOS tends to precipitate slightly more than REMO-FL. Based on the results in this and previous sections, we chose to use REMO-FL as the main source for analysis in the following sections, thus skipping
the other versions.

## 4   Lake analysis

We have chosen to analyze only big lakes due to the resolution of the simulations. This means that all lakes analyzed extend to at least 2 gridboxes (limits for the analyzed areas are shown in Table 2). When computing mean values for a specific lake, values for the gridboxes covering this lake were weighted by the lake fraction in these gridboxes (weighted arithmetic mean).
This way, the whole lake could be accounted for, even if the lake only covers a fraction of a specific gridbox. This approach causes some error to our analysis as parts of other lakes in the analyzed gridbox might influence the results, but since the model's resolution is relatively high and the number of these cases is small, we consider this to be a realistic approach for the lake analysis. If the resolution were to be increased, this method could eventually be replaced by direct analysis of the gridboxes covering a lake. We also analyzed the variance from the 35-year averaged results for all variables and it was found
to be so small that we have excluded it from the shown results.

### 4.1   Finnish lakes

We have analyzed the measured and modelled LWTs, ice thicknesses and snow depths on ice from 10 lakes in Finland. The measured snow layer thickness is calculated from the snow water equivalent (SWE) using the same density formulation as is used in the FLake (Mironov, 2008). This way, the results are more comparable and the actual thicknesses can be used instead
of water-equivalents. More detailed information about the measurement locations can be found from Table S1 (supplementary) and Fig. 1.

Figure 4 shows the results for all Finnish lakes included in this study. Overall, the main features are captured, but spring/early-summer LWTs are 3-10 °C lower in the model than what is measured and the modelled summer time maximum are overestimated by 2-3 °C at eastern lakes. Spring underestimations can be linked to the model's overestimation in the ice sheet thickness
due to the cold temperature bias shown in Sec. 3. It should be noted that the measurements are done close to the lake's shore, which means that measured LWTs are warmer during spring than the open-lake mean surface water temperature (which the modelled values correspond to). The underestimations in the model during spring can be mostly explained by these factors.





Summer time maxima are overestimated at lakes Pielinen, Kallavesi and Haukivesi. One explanation can be that these lakes are shallower in the model than in reality (Table 2) and they heat up too fast. This would also explain why the spring LWTs are first underestimated, but later on overestimated: the cold bias delays the ice break-up in the model, but as soon as the open-lake season starts, these shallower lakes start to heat up faster than in reality. Another reason for the overestimation can be that

these lakes have quite significant incoming and outgoing river discharge. This will cause some error when comparing LWT measurements with a lake model without river discharge. Otherwise the summer maxima are well captured, which indicates that although the annual open-lake time period is overall not as long as in the measurements, the summer is long enough for the lakes to warm to near-observed maximum temperatures. Autumn temperatures and features are well captured at all lakes by the model, although a small underestimation (1-3 °C) can be seen. The near-shore measurement location means that the

measured autumn LWTs are actually colder than the mean lake mean surface water temperature, because shallower shores cool faster than the open lake areas. This indicates that the measured autumn temperatures are actually a bit more underestimated in the model than shown in Fig. 4.

The ice thickness is reasonably well captured at most of the analyzed places, although the modelled values tend to be on the upper end of the measured ones. The measured values have wider spread, because multi-year daily means are shown and

the measurement day varies from year to year. This causes the measurement data have the wider spread, but it also gives some information about the variation in the measured thicknesses. As was discussed earlier, the model has a very small variance in the means, which means that the means shown in Fig. 4 are very representative for the whole period. Thus, it becomes clear that the model has a tendency to overestimate the ice thickness. The areas with biggest cold bias in Fig. 2 can be linked to the lakes with the highest ice thickness overestimation (up to 20 cm) in Fig. 4 (Haukivesi, Saimaa, Näsijärvi and Päijänne; in

the south and east part of Finland). The overestimation is evident also in the Table A1 values (appendix), where the measured mean ice-freeze-up and break-up times and ice period lengths are compared with the modelled values. Table A1 shows that the model tends to form ice 2-3 weeks too early, captures the ice-break up period reasonably well (1-2 weeks too late) and as a sum of these, overestimates the ice period length typically by three weeks. Naturally, there are no thickness measurements for the beginning or end of the ice season as the ice is too shallow to make these manual measurements. The measurement

location also influences the measured ice thickness. One example is Saimaa, where the measurements are done close to the Saimaa Canal. This shortens the measured ice season by delaying the start and advancing the end of it. Moreover, the definition of the actual date when a lake freezes or ice cover breaks-up is not straightforward. For example, during spring the shores can be ice free, but the open areas are still covered with ice. In the model, the ice season ends when there is no ice left in the whole analyzed area.

The amount of snow on ice seems to accumulate slightly too fast in the model, leading to an overestimation in the winter maximum values. As with ice thickness, the model mean values are closer to the measured maxima. Besides the model precipitation biases, which are small during winter (Fig. 2), there are a few physical reasons for the overestimation. Like with the ice thickness, measurements have more variability, whereas the modelled means are representative for the whole period (small variance). Wind is one factor that influences the snow layer thickness and its natural variability at all lakes. As FLake is a 1-D

model, there is no wind-driven transport of snow between the gridboxes and transport of snow is not accounted in REMO either.



This missing process can increase the modelled snow layer thickness quite significantly. This is especially important for more windy locations, such as lakes Lappajärvi and Pyhäjärvi. Both of these lakes are located in the western part of Finland, which is very flat and due to the proximity of Gulf of Bothnia also windy (see the orography in Fig. 1). Another factor influencing snow thickness is the lake size. While in FLake all snow, even if falling on a thin ice layer, will accumulate over ice, in reality,

the precipitated snow usually first forms a porous snow-ice layer before it starts to accumulate as pure snow. This happens especially over big lakes and thus, the modelled overestimation in the snow thickness can be partly explained by this process. On the other hand, this implies that the model overestimates the ice thickness even more than discussed above in connection to Fig. 4, because the measured ice thicknesses include both black ice and porous ice. Moreover, if we look at the Fig. S3 in the supplementary material where the measured and modelled snow amount on ice as well as the amount over land near lakes are

shown, we can see that the modelled snow thicknesses over lakes are roughly between the measured thicknesses over ice and land. Lakes Lappajärvi and Pyhäjärvi make an exception as at these lakes the modelled snow thicknesses are close to the ones measured over land. This suggests that the main reason for the overestimated snow thickness at these lakes is the missing snow transport due to wind. Naturally, there is some transport also over land areas, but it is not as efficient as over lakes, because the land surfaces are usually not as flat as the lake surfaces leading to less efficient snow transport over land. Nevertheless,

supplementary Fig. S3 shows that the modelled snow thicknesses on ice are not unrealistically high as they are within the measured snow thicknesses over lakes and land. In summary, we have identified three factors contributing to the overestimated snow thickness on lakes: the missing porous ice and snow transport processes, and the too early ice season, which allows snow to accumulate longer.

## 4.2  Swedish, Russian and Estonian lakes

Lake temperatures in Sweden, Russia and Estonia are well captured by the model, as can be seen from Fig. 5. The model shows a small tendency to overestimate the lake temperatures, especially during spring. On the other hand, the autumn temperatures are well captured. The annual cycle is reproduced at all locations, although at lake Onega the summer peak is roughly 5 °C too high. Overall, the model gives realistic LWTs at all analyzed locations.

Figure 5 shows that the model overestimates the ice period length on all locations, except at Võrtsjärv, where it is captured

very well. At other locations, the start of the ice period is 1-2 months too early and the ice cover lasts roughly one month too long. The ice period definition issues are one part of the problem, but if we look at how much there is still modelled ice when the measurements show the end of the ice period, we can see that the values are still reasonably high (10-40 cm). This tends to indicate that the main problem is the cold bias in by the model, especially when taking into account the too early start of the ice season.

The ice period length analysis includes some error due to the definitions problems already discussed in the previous section (how to define when a lake is ice free). In addition, the cold temperature bias over the large area of the domain increases the length of the modelled ice periods. The bias has an impact especially over the lakes in Russia. Some of the error is also coming from the measurements. For example, for the lake Vättern, the ice period start time was calculated by using the mean of only two years (1981 and 1984; there were no other data for the years 1981-1985) and the start time differed in these two years by





a month. In addition, the timing of the ice period can be quite difficult and this difference could be an explaining factor for differences in spring LWTs. One would expect that due to the earlier end of the ice season in the measurements, the LWTs would be higher in the measurements than in the model results, but this is generally not the case; rather, REMO-FLake even tends to overestimate the LWTs. Also, the measurement location can also play a role here; if it is close to the lake's shore,

the modelled spring values will be underestimated and autumn time overestimated. However, there is no clear signal like this, which would indicate that the measurement location is not a crucial factor here.

### 4.3   Vertical profiles from specific lakes

We have also compared the vertical profiles of LWTs. The analysis has been done only for 5 Finnish lakes, because measurement data were not available for all the locations analyzed in Sec. 4.1 (see Table S1 from the supplementary material).

Measurements are done during summer from a boat and winter through ice. The frequency of measurement data is usually 3 times per month and has some gaps during thin ice periods. This makes the measurement data frequency quite coarse compared to model data output frequency. Thus, we have filtered the model results to match the same dates as when the measurements were made. Additionally, as the measurements have some gaps, we have excluded from the analysis all mean values which are based only 1 or 2 measurement data points, to avoid the artificial inflation of the weight of sparse data points.

The calculation of the modelled vertical profiles is done by using the shape factor, mixing layer depth, mixing layer temperature and bottom temperature (details of these variable and the calculation method can be found from Mironov (2008)). During the ice period, FLake does not change the mixing layer depth; instead the last value before the ice period is used for the whole ice season. In the analysis, we have set the mixing layer depth to zero during the ice season and otherwise used the modelled depth. The mean depths of the lakes are similar in the model as in reality (see Table 2), but the maximum depths are

not available from Choulga et al. (2014). Thus, in Fig. 6 the measurement depth and model lake depth are on separate y-axes. In this way, we can compare the shapes of the temperature profiles while also looking into the actual temperatures. However, as the depths differ quite significantly, the profiles only tell how well the model captures the measured values in the current depth setup. With more realistic depths, the profiles could change and this error source should be kept in mind.

The vertical profiles in Fig. 6 have been averaged seasonally. The overall shapes of the temperature profile during winter

is captured by the model, but there is some underestimation of the deeper temperatures. In this respect, lake Päijänne makes an exception as there REMO-FL shows higher temperatures close to the lake bottom than the measurements. The use of deep soil temperature for lake bottom sediment temperatures can cause error, which could explain the differences near lake bottoms. Nevertheless, the overall difference in temperature during winter is only up to 2 °C in any of the lakes. During spring the profiles differ more than during winter. The near-surface temperatures show a larger discrepancy, which probably results again from

the model's cold bias: it prolongs the ice season during spring, thus lowering the near surface temperatures. The temperature profile at lake Päijänne is influenced quite a lot by the cold bias impacts and they shift the profile from the measured warmer top and colder bottom to the modelled colder top and warmer bottom. The spring difference in near-surface temperatures can be seen also from Fig. 4. At other locations, the modelled profiles are closer to the measured ones and overall the temperature difference stays within a few degrees.





During summer the profiles are fairly well captured. The measured and modelled surface temperatures are within 1-2 °C at all locations, but the near-surface temperatures differ at all locations. This is most probably caused by the assumed-shape representation of the temperature profile in the model combined with different depths between the model and measurements. Also, the near-bottom temperatures have a cold bias, especially at lake Pyhäjärvi, where it is almost 10 °C. This indicates

that the bottom sediment temperature has some error, a 35-year long spin-up run was performed. The bottom temperatures are mainly lower in the model than in measurements at all locations and all seasons. This is somewhat surprising as the lakes are shallower in the model than in reality and thus, one could expect higher heat transfer from the atmosphere to the lake bottom in the model during open-lake seasons. It is possible that the initial bottom temperatures, i.e. the deep soil temperature taken from the ERA-Interim data, were too cold and the spin-up time was too short to correct this. However, we see no drift in the

lake bottom temperatures which would suggest that this is not the reason.

The shapes of the temperature profiles during autumn are in reasonable accord with the measurements. The measured values are higher than the modelled ones at all locations. The near-surface values are within a couple of degrees, but the near-bottom temperatures differ by up to 5 °C. However, despite the differences and uncertainties related to analysis coming from the different lake depths, the seasonally averaged vertical profiles are realistic and show that the REMO-FLake has the capability

to reproduce the vertical profiles of the lake temperatures.

## 5   Conclusions

In this work, the regional climate model REMO was interactively coupled with the lake model FLake (REMO-FLake). With the new version we have simulated the Fenno-Scandinavian climate over 35 years and evaluated the model in terms of climate and lake-related variables. Fenno-Scandinavia has a large number of lakes of various sizes, making it a very suitable domain

for a coupled regional-lake model. In addition, we have tested how sensitive the model is to different lake parameters and how much the snow albedo scheme influences the winter time climate.

REMO-FLake can reproduce the Fenno-Scandinavian climate realistically. However, the REMO model tends to have an overall cold bias over northern areas when there is snow on the ground. This is also visible in REMO-FLake, which in fact enhances the cold bias in winter. The reason for this is that the standard model version gets the lake temperature and ice cover

from the nearest sea-point, leading to unrealistic heat and moisture sources, and thus decreasing the underlying bias problem. Excluding the snowy seasons, REMO-FLake captures the Fenno-Scandinavian temperatures better than the original version. In terms of precipitation, REMO-FLake outperforms standard REMO in all others seasons than summer. During summer, the convective precipitation is too active in REMO-FLake leading to a wet bias over areas with a high lake density.

We analyzed in detail the lake water temperatures (LWT), ice thicknesses and snow amounts over ice from 10 different

Finnish lakes. The results show that the model can capture the LWTs well, although there are some differences during spring due to the longer modelled ice period associated with the cold bias. The ice thicknesses tend to be overestimated, especially in areas where the cold bias is strongest. The snow thickness on ice also shows slight overestimation, which is caused either by the missing porous ice formation in FLake or the missing horizontal snow transport due to wind. Overall, the model performs

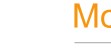

well in terms of the variables analyzed. We also did some LWT and ice depth analysis for lakes outside Finland. This analysis showed that the model performs very well throughout the Fenno-Scandinavian domain. Ice depth and ice season length had some overestimations due to the cold bias in the model, but overall, the differences to measurements were small. The vertical lake temperature profiles were also analyzed and they showed that the modelled profiles have very similar features as the

measured ones, while the modelled near-bottom temperatures are generally underestimated.

We did not analyze in detail the reasons for the cold bias in this work. However, we did test how sensitive the model is to the snow albedo scheme, which is originally based on a snow temperature approach. We implemented another scheme originating from the Biosphere-Atmosphere Transfer Scheme (BATS), which takes into account the aging of snow, soot loading, grain size and the influence of solar zenith angle. Our test simulations shows that the snow albedo has only a minor impact on the

cold bias. Although the snow albedos had differences between the two schemes, the impact on simulated climate is reduced by the rather low intensity of solar radiation during the snow season in Fenno-Scandinavia. A detailed analysis of the cold bias problems and causes behind it will be left for forthcoming publications. From a technical point of view, the computational costs of the implemented lake model are insignificant, thus further suggesting the use of the new model version.

## 6  Code availability

The source code for FLake is publicly available (http://www.flake.igb-berlin.de/). The sources for REMO(-FLake) are available on request from the Climate Service Center Germany (contact@remo-rcm.de).

## 7  Data availability

Due to the very large size of the data files, the data are not publicly available, but they can be requested from the first author.

*Author contributions.*  J.-P. Pietikäinen did the lake model coupling based on earlier work by J. Kaurola and the coding work for lake tiles

and snow albedo. J.-P. Pietikäinen also planned the simulations, did major parts of the analysis and wrote the manuscript. T. Markkanen and Y. Gao helped with the connections between the main model and the lake model/tiles (in terms of soil and snow). K. Sieck and D. Jacob assisted with the main model structure (tiles) and physical processes. Johanna Korhonen gave her expertise on water temperature and ice and snow thickness measurement data. P. Räisänen gave his expertise on the snow albedo part and J. Ahola assisted with the lake temperature analysis. H. Korhonen and A. Laaksonen assisted in the analysis and manuscript writing phases and A. Laaksonen also initiated the work.

The coauthors also helped with the manuscript by giving valuable comments.

*Competing interests.*  The authors declare that they have no conflict of interest.





*Acknowledgements.* The authors would like to thank Pentti Pirinen and Juha Aalto from the Climate Service Centre of the Finnish Meteorological Institute for providing the gridded data over Finland and helping with technical aspects. We are grateful to Margaret Choulga from the Russian State Hydrometeorological University for providing the updated lake data. The insights about the FLake model by Dmitrii Mironov from the Deutscher Wetterdienst have been very valuable and the authors would like to express their gratitude. We also want to thank Aku

5  Riihelä from Finnish Meteorological Institute for his help with the satellite albedo measurement data. We are also thankful for the insights regarding the snow albedo from Thomas Raddatz from the Max Planck Institute for Meteorology. We acknowledge the E-OBS dataset from the EU-FP6 project ENSEMBLES (http://ensembles-eu.metoffice.com) and the data providers in the ECA&D project (http://www.ecad.eu). Hannele Korhonen acknowledges the funding from the European Research Council (ERC) under the European Union's Horizon 2020 research and innovation programme under grant agreement No. 646857.

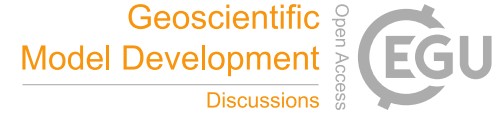

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



**Table 1.** Simulation names with the configurations showing whether FLake is used or not, which snow albedo scheme ($\alpha_{sn}$) is used and which snow heat conductivity ($k_s$) method is used.

| Simulation | FLake | $\alpha_{sn}$ | $k_s$ (lakes) |
|---|---|---|---|
| REMO-ST | No | T-scheme | - |
| REMO-FL | Yes | T-scheme | Semmler et al. (2012) |
| REMO-FLB | Yes | BATS | Semmler et al. (2012) |
| REMO-FLTB | Yes | T-scheme + BATS | Semmler et al. (2012) |
| REMO-FLOS | Yes | T-scheme | FLake original |

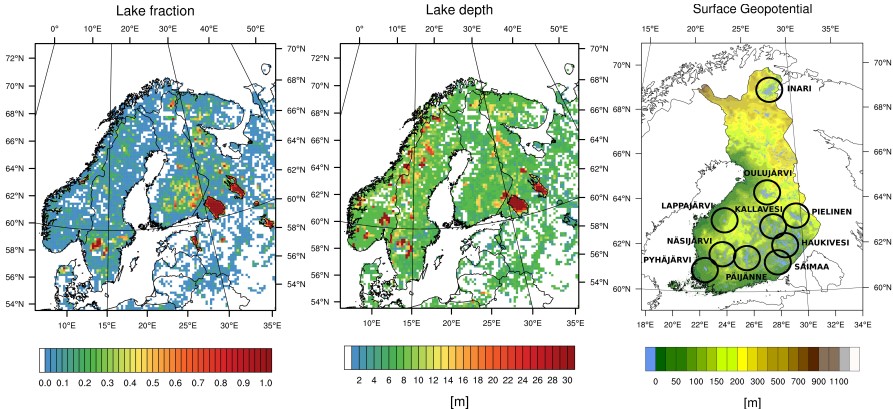

**Figure 1.** Lake fraction (left, based on GLCC data) and lake depth (center, based on Choulga et al. (2014) data) of the calculation domain. Additionally, the surface geopotential height of Finland and the locations and names of the analyzed Finnish lakes (right, based on Aalto et al. (2016) data).

## Appendix A: Ice cover period analysis





**Table 2.** Analyzed lakes with their mean depth, mean depths applied in REMO and the lake area information for the REMO analysis (SYKE refers to the Finnish Environment Institute).

| Lake | Data source | Mean depth [m] | REMO area mean depth [m] | REMO area longitudes | REMO area latitudes |
|---|---|---|---|---|---|
| Haukivesi (Fin) | SYKE | 9.1 | 7.0 | 27.9 ↔ 28.9 | 61.9 ↔ 62.3 |
| Inari (Fin) | SYKE | 14.3 | 13.3 | 27.0 ↔ 28.6 | 68.7 ↔ 69.2 |
| Kallavesi (Fin) | SYKE | 9.7 | 7.0 | 27.4 ↔ 28.0 | 62.5 ↔ 63.0 |
| Lappajärvi (Fin) | SYKE | 6.9 | 16.0 | 23.4 ↔ 23.9 | 63.1 ↔ 63.3 |
| Näsijärvi (Fin) | SYKE | 13.7 | 7.0 | 23.5 ↔ 23.9 | 61.5 ↔ 61.9 |
| Oulujärvi (Fin) | SYKE | 7.0 | 6.8 | 26.7 ↔ 28.0 | 64.1 ↔ 64.6 |
| Pielinen (Fin) | SYKE | 10.1 | 9.8 | 29.0 ↔ 30.2 | 62.9 ↔ 63.6 |
| Pyhäjärvi (Fin) | SYKE | 5.5 | 5.4 | 22.1 ↔ 22.4 | 60.9 ↔ 61.1 |
| Päijänne (Fin) | SYKE | 14.2 | 14.5 | 25.1 ↔ 25.9 | 61.2 ↔ 62.2 |
| Saimaa (Fin) | SYKE | 10.8 | 13.6 | 27.7 ↔ 28.8 | 61.1 ↔ 61.6 |
| Hjälmaren (Swe) | ILEC (1988-1993) | 6.2 | 5.9 | 15.3 ↔ 16.3 | 59.1 ↔ 59.3 |
| Mälaren (Swe) | ILEC (1988-1993) | 13.0 | 6.1 | 16.1 ↔ 17.1 | 59.2 ↔ 59.6 |
| Vänern (Swe) | ILEC (1988-1993) | 27.0 | 22.0 | 12.3 ↔ 14.1 | 58.4 ↔ 59.4 |
| Vättern[1] (Swe) | ILEC (1988-1993) Benson and Magnuson (2000, updated 2012) | 41.0 | 29.0 | 14.1 ↔ 15.0 | 57.8 ↔ 58.8 |
| Ladoga[1] (Rus) northern part | ILEC (1988-1993) | 51.0 | 62.4 | 29.8 ↔ 33.0 | 60.8 ↔ 61.8 |
| Ladoga[1] (Rus) whole lake | Benson and Magnuson (2000, updated 2012) | 51.0 | 47.3 | 29.8 ↔ 33.0 | 59.9 ↔ 61.8 |
| Onega[1] (Rus) | ILEC (1988-1993) Benson and Magnuson (2000, updated 2012) | 30.0 | 25.7 | 34.3 ↔ 36.5 | 60.9 ↔ 62.9 |
| Võrtsjärv[2] (Est) | ILEC (1988-1993) | 2.7 | 3.0 | 25.9 ↔ 26.2 | 58.1 ↔ 58.4 |

[1] lake temperature from ILEC (1988-1993) and ice cover period from Benson and Magnuson (2000, updated 2012)

[2] lake temperature and ice cover period from ILEC (1988-1993)



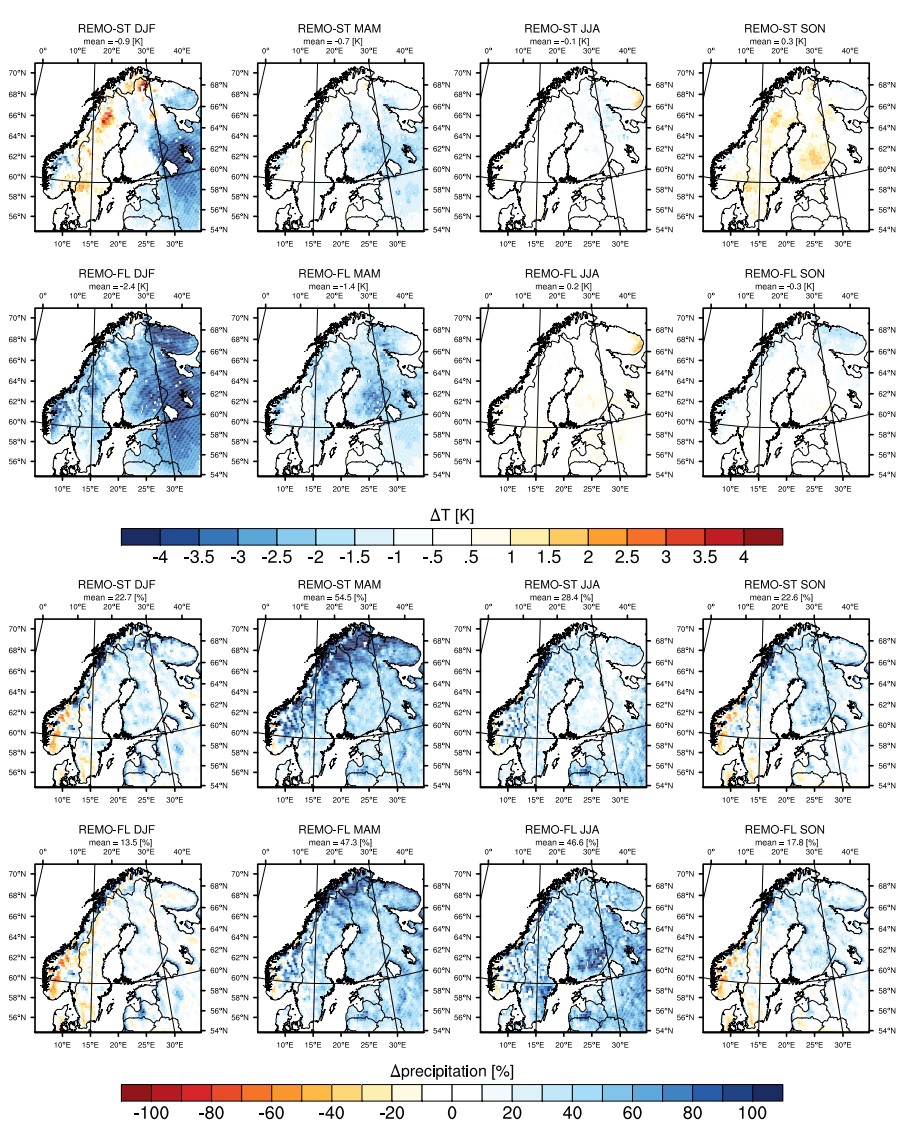

**Figure 2.** 2-m temperature and precipitation biases of standard REMO (REMO-ST) and REMO with the FLake module (REMO-FL) compared to E-OBS data. The seasonally averaged results are for the time period of October 1979 - March 2015.

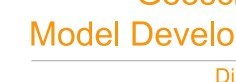
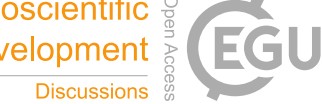


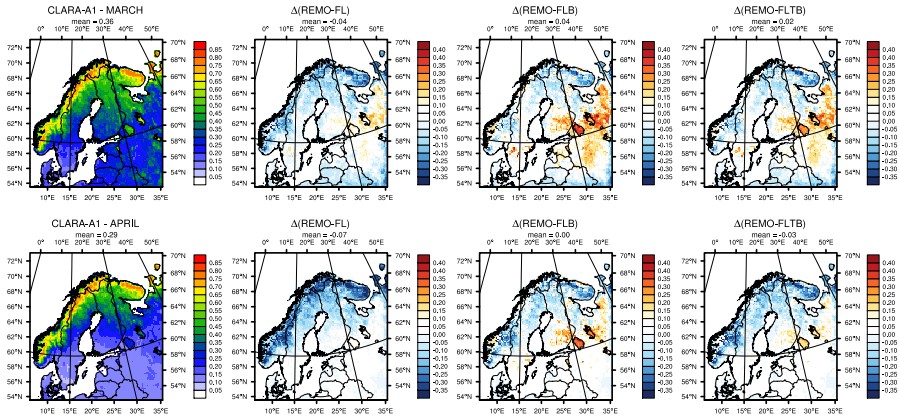

**Figure 3.** Monthly mean albedos for March and April from satellite data (CLARA-A2; Karlsson et al., 2017) and the corresponding bias (model-measurements) in the REMO-FL, REMO-FLB and REMO-FLTB runs. The monthly means are for the time period of 1982-2014 and only for land and lake surfaces.

**Table A1.** Measured ice-period start and end days (as the day of the year) and the length of the ice period in days. The differences to modelled values are also shown (model−measurements, i.e. when the values are negative, the model is too early in its prediction and when the values are positive, the model is late in its prediction). If the difference between measured and modelled ice-period start, end or length is less than two weeks, the values are boldfaced. The values have been averaged for the time period of 1979-2015.

| | measured ice-period start | model difference | measured ice-period end | model difference | measured ice-period length | model difference |
|---|---|---|---|---|---|---|
| Haukivesi | 339 | -15 | 127 | **7** | 154 | 22 |
| Inari | 313 | **-13** | 151 | **6** | 204 | 19 |
| Kallavesi | 340 | -17 | 130 | **5** | 156 | 22 |
| Lappajärvi | 330 | **3** | 126 | **7** | **162** | **4** |
| Näsijärvi | 350 | -22 | 123 | **3** | 139 | 25 |
| Oulujärvi | 326 | -17 | 138 | **5** | 178 | 22 |
| Pielinen | 330 | **-14** | 134 | **8** | 170 | 22 |
| Pyhäjärvi | 342 | **-9** | 116 | **5** | **140** | **14** |
| Päijänne | 347 | **-13** | 123 | **12** | 142 | 25 |
| Saimaa | 344 | -17 | 119 | **11** | 141 | 28 |





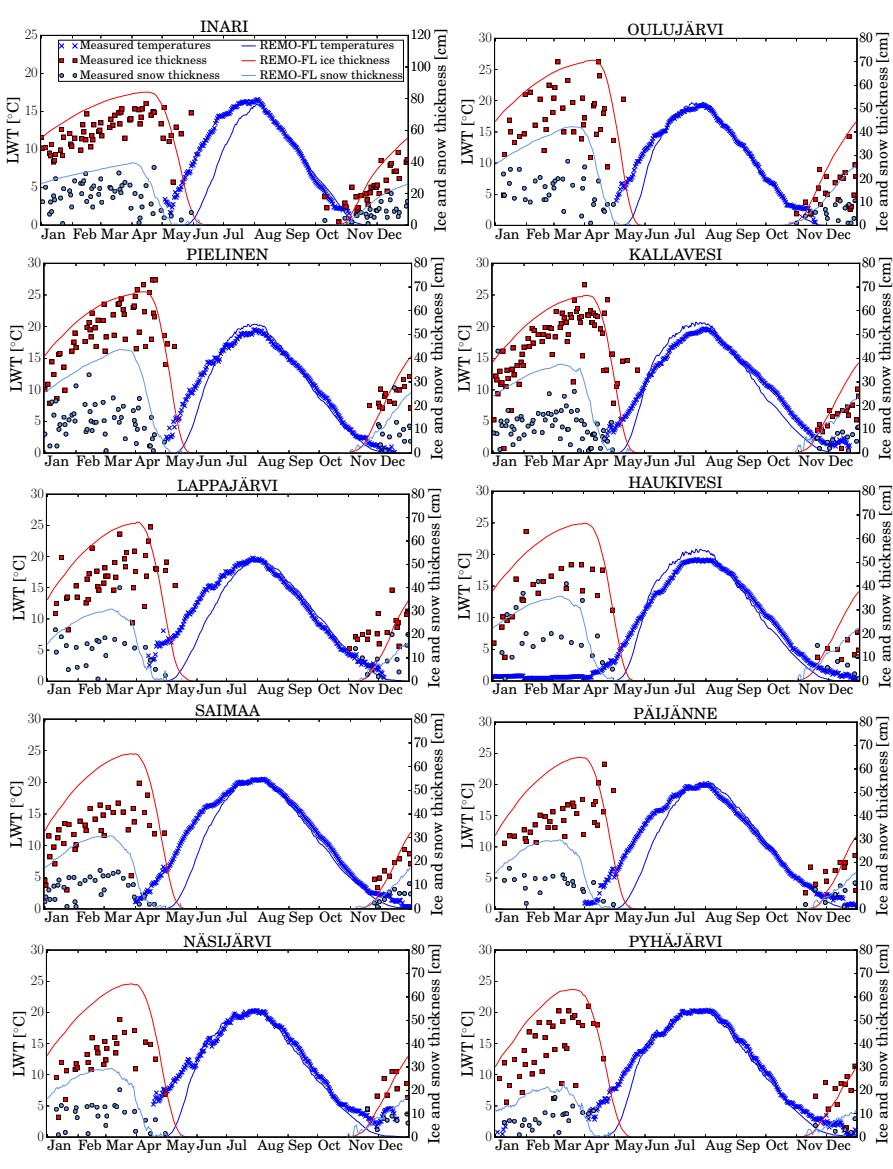

**Figure 4.** The measured and modelled daily averaged lake water temperatures (LWTs), ice thicknesses and the amount of snow on ice. The values have been averaged for the time period of October 1979 - March 2015.





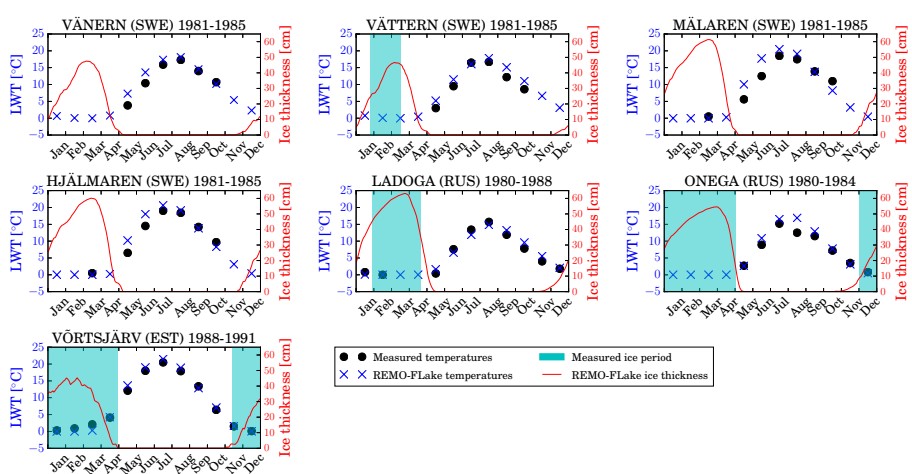

**Figure 5.** Measured and modelled lake water temperatures (LWTs) and ice periods from Sweden, Russia and Estonia. The lake temperatures for Lake Ladoga are calculated only for the northern part (following the measurement approach), but the ice period and depth are for the whole lake.



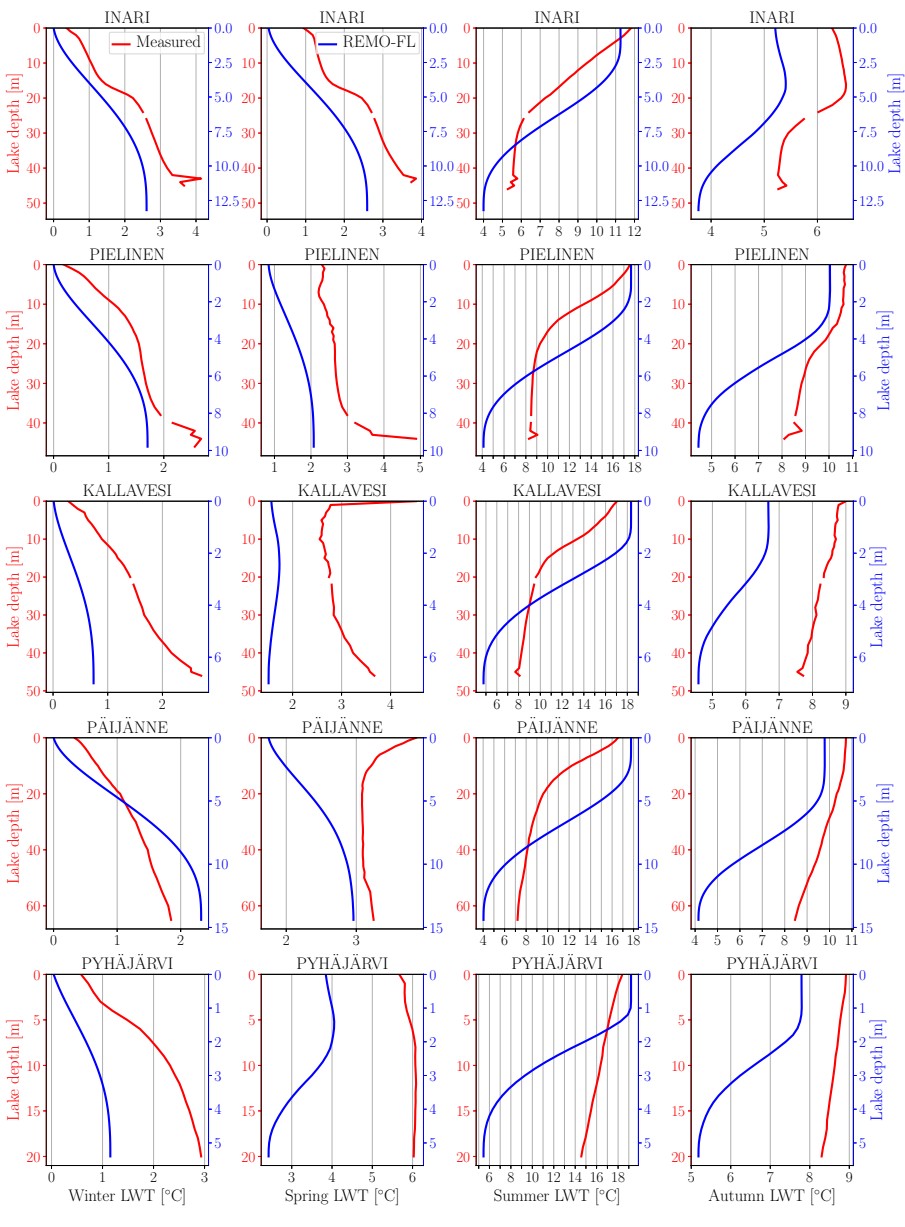

**Figure 6.** Vertical profiles of the measured and modelled seasonally averaged lake water temperatures (LWTs) in different lake depths. The values have been averaged for the time period of 1979-2015. The lake depth is shown for measurements (red color) and model results separately (blue color). Also, we have averaged the model results only over those days when measurements were available, because the measurement frequency was low (a few times per month).