# Peer review of "The regional climate model REMO (v2015) coupled with the 1-D freshwater lake model FLake (v1): Fenno-Scandinavian climate and lakes"

_Geoscientific Model Development, 2017_

## Short Comment (SC1) · 13 Dec 2017

As explained in https://www.geoscientific-model-development.net/about/manuscript_types.html GMD is expecting that authors upload the program code of models (including relevant data sets) as a supplement or make the code and data citable through a DOI (digital object identifier) for the exact model version described in the paper. As the FLake is "freely available" the authors should apply one of these access options.

If for some reason the code and/or data cannot be made available in this form authors need to state the reasons for why access is restricted. As REMO(-FLake) is available on request only you need to clearly state why and how the access is restricted (in

general this would be for licensing reasons).

Lutz Gross GMD Executive Editor

---

## Referee Comment (RC1) · P. Le Moigne (Referee) · 14 Dec 2017

The comment was uploaded in the form of a supplement:
https://www.geosci-model-dev-discuss.net/gmd-2017-245/gmd-2017-245-RC1-supplement.pdf

---

## Referee Comment (RC2) · Anonymous Referee #2 · 14 Dec 2017

**General comments**

The manuscript presents a study of the impact of a lake model, FLake, in the regional weather model REMO over several decades. Results are presented of mean seasonal biases over the study area, together with mean annual and seasonal evolution at a number of selected lake sites.

The main finding of the study is that FLake produces good lake surface properties, perhaps despite the characteristics of the model within which it is embedded. However, its inclusion exacerbates the worst existing model biases, and the modelled internal lake temperature structure may not be a good representation of the measured lake

temperature profiles.

In terms of its significance, it adds to a number of studies (not all cited here) validating the use of FLake in global or regional weather or climate models. Previous examples may be seen in the special issue of Boreal Environment Research (vol.15(2), 2010), and the Tellus thematic cluster "Parameterization of lakes in numerical weather prediction and climate models".

**Specific comments**

*Section 2.1*
Given the apparent model biases, and later error attribution, the description of the boundary-layer and surface-exchange representation of REMO is inadequate. Of the references given (p3, line 22), Kotlarski (2007) is a PhD thesis and Rechid (2009) is no longer available at the URL cited. In any case some description of the main points should be given here rather than relying solely on references.

The description of tile structure is also confusing. It is stated that there are 3 tiles: land, ice and water (p3, line 23), but then that the standard land-surface scheme does not have a tile for lakes (p3, line 25). So does "water" mean "sea water", as on p4, line 15?

*Section 2.3*
P4, line 19: The GLCC lake fractions were used instead of those from Choulga et al (2014), although the latter are consistent with the depths taken from the same dataset. How do the two lake-fraction datasets compare?

P4, lines 25–26: The fractional ice-cover in the original model is replaced with binary ice cover when FLake is implemented. It is possible that this change on its own may affect the model results. Its impact should have been estimated.

P5, lines 8–15: There is some discussion of heat conductivity for lake ice and snow on lakes, but no discussion of how snow on land is modelled. Snow heat conductivity is taken as a single number (0.14 W/(mK), p5, line 14) despite the fact that snow density

is apparently not fixed (p5, line 9). Studies such as Calonne et al. (GRL 2011) show that conductivity varies greatly with snow density, and the weighted sum of equation (1) presumably acts partly as a substitute model for this variation. It may be speculated that over-simplistic representation of snow conductivity generally may contribute to the winter cold bias even in the control run.

*Section 2.4*
Snow albedo gets a subsection here, although the rest of the snow scheme is not described. As stated later (e.g. p10, line 33), albedo changes may have little effect in winter given the lack of solar radiation in these regions then, so the emphasis on this aspect needs more justification.

P5, line 29: The forest fraction mentioned here should have been described earlier in section 2.1.

P6, lines 3–11 seem to be a repeat of the previous paragraph.

*Section 2.5*
P6, line 32 – p7, line 2: The model end-state (in March?) is used as the model initial state (in August?). Would it not have been more suitable to use an initial state from the same season, say from August of the penultimate year?

P7, lines 5–6: Given that one comparison is with 2 m temperature, the height of the lowest model level should be stated, as also the surface interpolation scheme should earlier have been described.

*Section 2.6*
P7, line 31 – p8, line 1: Apparently 7 out of the 18 chosen study lakes may have unrepresentative mean depths in the model dataset. However, lake depth is often described as the parameter to which FLake is most sensitive. Is the study intended to test, or contradict, this assertion?

*Section 2.6*

[Figure]

P 8, line 32. "The reason for the [control-run] cold bias is still unknown." Given that on p2 (lines 7–9) it is anticipated that a better lake representation will reduce wintertime heat fluxes, it seems odd to prioritise implementation of a lake model into a weather model with an existing, unexplained winter cold bias. Not least, it undermines the ice thickness/duration comparison. (As an aside, does the cold bias correlate with snow depth, as hinted by p9, lines 4–5?). The neglect of the cold-bias problem is acknowledged, but not really explained, in the last Conclusion paragraph.

As a related point, the discussion on p11, lines 3–7 seems to say that the snow conductivity has been adjusted over lakes only, but acknowledges that snow thermodynamics may be a large factor in the cold bias. So why not try and improve "land" snow behaviour before applying FLake?

*Section 4.1*
P11, lines 27–32: "The underestimations in the model during spring can be mostly explained by these factors." The factors are discussed in a rather vague and unquantitative way, hence the proposed explanations are not very convincing.

P12, lines 1–6: Errors are attributed to having the wrong mean lake depth, or to river input/output. It might have been better to adjust the depths of the study lakes, at least, to try and differentiate between these possibilities. Not least, it would be interesting to definitely attribute errors to river flow (or not), since this has a bearing on the general usefulness of 1-D models with no hydrological connections.

P12, lines 23–24: "Naturally, there are no thickness measurements for the beginning or end of the ice season as the ice is too shallow to make these manual measurements." Does this mean the thickness measurements are biased high?

P12, line 30 – p13, line 18; Again, the reasons for discrepancies are discussed in a qualitative way, with no quantitative testing or discrimination between the possible causes of error advanced.

[Figure]

*Section 4.2*
Figure 5: It appears that the surface temperature at Vörtsjärv is above freezing during some or all of the ice-covered period?

P13, line 28: "the main problem is the cold bias [of] the model." I believe this supports my earlier point about priorities.

*Section 4.3*
Figure 6: I do not find the different (arbitrarily chosen?) depth scales for model and measurement profiles to be a useful way of making this comparison. Instead, profiles with a common depth scale should be shown. Then the authors can discuss (or the reader can decide) whether FLake gets the correct surface behaviour despite, or because of, its internal lake-structure model.

**Technical points**

Title: Should it be "freshwater **lake** model FLake"
p1, line 10: "are in realistic." ?
p3, line 8: use NWP as defined on p1.
p4, line 11: Remove "on"
p5 line 1: Does "It" refer to FLake?
Equation (1): $hs$ does not seem to be defined.
p5, line 19: tp -> to
p10, line 27: tot he -> to the

---

## Author Comment (AC1) · 2 Feb 2018

Dear Executive Editor Gross,

thank you for your comment. FLake model is freely available from FLake's web-page and it is under the MIT license. The model source code as well as the input data can be freely downloaded from the web, although the developers ask all user to contact them so they know who is using the model. We have stated in the manuscript which version of the model code and input data we are using and these can be downloaded from the FLake web-page (keeping in mind that the license should be read and users

are asked to contact the developers).

REMO-FLake has some minor code changed also to FLake part (interface coupling). REMO(-FLake) itself is availabe from the Climate Service Centre Germany, but all users have to sign a license agreement.

We will modify the Code availability sections to be more clear on these aspects.

Best Regards,
Joni-Pekka Pietikäinen

---

## Author Comment (AC2) · 2 Feb 2018

**General comments:**

**This paper describes how the regional climate model REMO was coupled with the 1D lake model FLake, and what were the impacts on regional climate over the Fenno-Scandinavian region. For that purpose, a specific tile was added to the existing ones to represent lakes, and several experiments were conducted first to assess the impact of using FLake coupled to REMO and then to study the sensitivity of the coupled REMO-FL model to the snow albedo parameterization. These simulations were evaluated against in situ measurements for lakes and**

[Figure]

**regional datasets for the coupled system. The main conclusions are that the coupled system improves the realism of simulations but enhances an already existing cold bias. There's a high interest in understanding lakes behaviors for their coupling to the atmosphere, especially in a context of climate change and the effort made to couple REMO to FLake is very valuable. Moreover, regional climate models tend to increase their horizontal resolution where surface is better defined. In regions with a high lake density, more and more small lakes will be resolved and potentially will have an impact on the local and regional climate. The paper is well written and structured, and a substantial effort was performed in the evaluation of the coupled model. I found it very interesting and easy to follow. Although the subject is of interest, I have noticed several aspects that needed to be more documented. Therefore, before accepting the manuscript for publication, minor modifications are required. These are listed below.**

We thank the reviewer for valuable comments for improving the manuscript. Throughout the text reviewers comments are marked with boldface and after each comment follows our reply. We also provide the page and line number(s) of the revised manuscript where the modification(s) can be found.

**Specific comments**

**Coupling: This is not clear how the coupling between the surface and the atmosphere is performed. I understand that there are tiles over which calculations are performed at each time step of the model. Then what are the quantities that are aggregated in the gridbox? Are surface fluxes or surface variables aggregated and then transferred as boundary conditions to the radiation and turbulence schemes?**

We will improve this part of the manuscript. The lake related variables (temperature, albedo, roughness length, latent and sensible heat flux, specific humidity etc.) are calculate tile-wise while the model physics mostly uses mean values over the grid cells (weighted by tile area fractions). Tile-wise variables are included in the diagnostic output of the model. Revised manuscript: P3L22 and forward.

**How many layers are used to represent the PBL? It may be important to well capture the PBL representation especially in such a work where surface is a key component and can affect processes like lake breeze, evaporation, low clouds formation, etc.**
It varies slightly during the simulations, depending on the PBL depth, but on average for this domain there are 6 layers describing the PBL. Overall, the vertical resolution is higher in the lower troposphere than higher up. Using more vertical levels should improve the results, but for our simulations we are confident that this would not change the results substantially. The choice in number of the vertical levels (and spatial resolution) has to be always balanced with the computational resources. We have added the information about the number of levels within PBL in the manuscript. Revised manuscript: P7L21-23

**Is there an implicit coupling between the atmosphere and each surface tile?**
The calculations are as following: during a time step all tile-related variables are calculated separately and later on grid box variables (2-meter temperature, humidity etc.) are calculated as a mean value weighted by the tile area fractions. Naturally, many physical routines (radiation, cloud scheme etc.) use these grid box variables and thus, during the next time step all tiles will get updated information (like radiation fluxes) that was based on tile averaged values. We will improve the description of this part. Revised manuscript: P5L11-14

**FLake:**

**The choice of using the REMO module to computed surface fluxes was not explained. This is not said if the SfcFlx module was tested in REMO and what were the results as compared to the REMO module. Was it at least tried and surface fluxes compared to some in situ measurements?**

We will improve this part. Shortly, we did not implement nor test the SfcFlx module, because we wanted the lake tile surface fluxes to be in equilibrium/consistent with the other tiles. A similar approach has been used in previous FLake-RCM/GCM coupling works. Revised manuscript: P5L15-16

**The authors decided to activate the bottom sediment for lakes having a real depth smaller than 50m. There are however no measurements available (or only a few) within the bottom sediment layer to validate this module and I'm not convinced that a 35-yr spin-up is enough to correctly initialize sediment characteristics. A drawback of the activation of this module could be an enhancement of a temperature bias at the lake bottom that could affect the whole temperature profile. What was the strategy to decide to use or not the sediment bottom module? What is the impact on the simulated temperature profiles?**

The original paragraph has some missing information and is not clearly written. We will improve this part and add the missing information. In short, the FLake webpage says "Apart from shallow lakes, the bottom heat flux can be set to zero... Experience suggests that for lakes deeper than about 5 m the heat flux through the bottom can safely be neglected." This was also the case with our implementation. Also, when FLake is used for deeper lakes, a 50 m "false bottom" is used for temperature profile and the bottom sediment is switched off. Naturally, already the first 5 m cut-off limit switched off the bottom sediment module, but we left the possibility to use it also for deeper depths than 5 m (as is the case with default FLake as well).

So the impact was negligible, because in Fig. 1 there were only few grid boxes with depths less than 5 m. Revised manuscript: P5L20-27

**Nothing is said on the light extinction coefficient which is another important FLake parameter. Which value was chosen for REMO-FL? Does it vary in space, in time? Depending on its setup it may have a non-negligible impact on surface temperature and consequently on surface fluxes during the ice-free periods, but also on lake temperature profiles.**
We are using the default configuration. There has been some papers about this and indeed they suggest using different values, depending on the locations. Overall, it seems that a global map for the light extinction coefficient(s) is needed and perhaps even a time-dependent approach could be utilized (Zolfaghari et al. (2017)). In this work, we have not tried to solve issues with this part of the FLake model. In terms of the manuscript, we will add information about using the default values for the light extinction coefficient. Revised manuscript: P6L9-10

Citation: Zolfaghari, K., Duguay, C. R., and Kheyrollah Pour, H.: Satellite-derived light extinction coefficient and its impact on thermal structure simulations in a 1-D lake model, Hydrol. Earth Syst. Sci., 21, 377-391, https://doi.org/10.5194/hess-21-377-2017, 2017.

**REMO:**

**In the Semmler approach for snow conductivity, the C coefficient is an empirical constant. It was setup for Bear Lake in Alaska. Is the formulation adapted for snow over lakes in the Fenno-Scandinavian region?**
We did no make any changes to Semmler's approach. It is an improvement to the default setup already as it is (others have done the same, also on the NWP side). It is

very probable that this approach could be improved with more local information.

**The snow albedo is limited to 0.25 in case the gridbox is forest-covered. How-ever, in Kolarski et al. 2007 (reference of Gao et al. 2014 in the manuscript that refers to it), the figure 3.6 shows that the lower limit is 0.3 and not 0.25. What has motivated the choice of using 0.25 instead of 0.3? Are there simulations showing that this particular value performs better in terms of snow thickness, temperature?**
The motivation was the literature based values (Roesch et al., 2001) which showed that the limits could be decreased. In this work we followed the Gao et al. 2014 ap-proach as was shown to work better for the domain used in this work (e.g. temperature was closer with measurements). We did also make a test simulations with the default REMO values and got similar results as Gao et al. 2014.

**Evaluation:**

**E-OBS was chosen as the truth for the evaluation of the coupled system. It's clearly mentioned that E-OBS relies on the observation density network, which is true. However, how is the comparison between observations and simulations for 2m-temperature performed: is the difference of orography accounted for in E-OBS and in the model? This can have a very strong impact in mountainous regions. On the other hand, climatology of CRU is usually used (this is not the only one) to evaluate screen variables in climate models. Even if the resolution is coarser than what E-OBS can provide as interpolator, it would have been worthwhile comparing the simulations to another dataset. Particularly, it would be interesting to know if E-OBS is able to well capture convective events since there is a large difference with the simulations.**
Orography influence should be always taken into account and we have also done it. This information was missing from the text and we have added it. CRU data would

have been another source (we have used it before), but in this work we concentrated on E-OBS. We also checked for Finland if using FMI's $10 \times 10$ km$^2$ resolution dataset would bring some more insight to the analysis, but the differences to E-OBS were small (even during summer when there are a lot of convective events over Finland). This, and the knowledge that the E-OBS observation network density is quite high over the Nordic countries, convinced us that using only E-OBS is more than sufficient for our study. Revised manuscript: P9L8-9

**As a complement, it would help the readers to know the impact of using REMO-FL on other near-surface fields like 2m-relative humidity or 10m-wind speed by comparing with REMOST. Because for example differences in humidity will have a direct influence on evaporation. Could you add a figure showing how lakes modulate these fields and add a discussion on this point?**
A good suggestion. We have added the requested figure and discussion to the analysis. Revised manuscript: Fig. 4, Sec. 3.1

**Discussion:**

**The authors mention page 8 that LWTs are usually close to T2M. This is not really true because it assumes near-neutral conditions which are unlikely to occur especially in winter when the lake is ice or snow-covered. In that case there is a decoupling between the surface and the air just above leading to very cold temperatures at surface. This decoupling may be responsible for the model cold bias. Can you add a comment on that?**
The original formulation was misleading. The aim was to say that during open-lake season (no ice) the LWT's can act as a proxy for the 2-m temperature and we will analyze these later. Here, the compared temperatures are the real 2-m temperatures from the model and there is no artificial bias coming from the method. We will improve

the text to avoid any further misunderstandings. Revised manuscript: P9L19-20

**In page 9, figure 2 shows only relative differences for precipitation between model and observations. The absolute value is lacking (a difference of 50% for a 100mm rainfall will not have the same impact than on a 10mm one). Same for temperature, the reference is lacking. Please add the figure S1 (E-OBS reference data) from your supplementary material in the manuscript. On top of this remark, are the differences significant (for instance, was a Student test performed)? If not it would be worth examining if all differences experienced in the domain are significant or not.**
We have added S1 to the main text and improved the figure a bit. Statistical tests were not done before, but we have now added this analysis and all the figures showing differences against E-OBS (also in the supplementary material) show now the statistical significance with p-values $< 0.05$ with black dots. Overall, it can be said that the differences are statistically significant. Revised manuscript: Figs. 2 and 3 + Sec. 3.1

**Technical comments**

**Regions 126, 130 and 132 are not explicit, please explain what they correspond to?**
This information was too detailed and we have changed the ending of the paragraph to: "For points where Choulga et al. (2014) has no data, a default value is used and it can be set beforehand within the surface pre-processor. In this work, a default value of 7 m was used, which is a typical mean depth for our domain based on the data by Choulga et al. (2014)."

**Page 6, there seems to have a redundant paragraph. "The radiation... 2014" should be removed.**
This is true → paragraph was removed.

**Page 6, line 17, repetition of "grain growth effect..."**
Corrected to "the first represents the effect on snow grain growth due to vapor diffusion"

**Page 10, typo: tot he → to the**
Corrected as suggested.

**Page 12, line17: "means" is used three times in the same sentence. You could replace for instance, the second one by "indicates".**
Corrected as suggested.

**Page 12, line15: data have → data to have**
Corrected as suggested.

**Page 14, line14: only → on**
Corrected as suggested.

**Figure 3 refers to CLARA-A1 whereas manuscript refers to CLARA-A2**
The figure title had old information (at an earlier stage of manuscript preparation, we used CLARA-A1 data). Changed to CLARA-A2.

---

## Author Comment (AC3) · 2 Feb 2018

**General comments:**

The manuscript presents a study of the impact of a lake model, FLake, in the regional weather model REMO over several decades. Results are presented of mean seasonal biases over the study area, together with mean annual and seasonal evolution at a number of selected lake sites.

The main finding of the study is that FLake produces good lake surface

properties, perhaps despite the characteristics of the model within which it is embedded. However, its inclusion exacerbates the worst existing model biases, and the modelled internal lake temperature structure may not be a good representation of the measured lake temperature profiles.

In terms of its significance, it adds to a number of studies (not all cited here) validating the use of FLake in global or regional weather or climate models. Previous examples may be seen in the special issue of Boreal Environment Research (vol.15(2), 2010), and the Tellus thematic cluster "Parameterization of lakes in numerical weather prediction and climate models".

We thank the reviewer for valuable comments for improving the manuscript. Throughout the text reviewers comments are marked with boldface and after each comment follows our reply. We also provide the page and line number(s) of the revised manuscript where the modification(s) can be found.

**Specific comments**

**Section 2.1**

Given the apparent model biases, and later error attribution, the description of the boundary-layer and surface-exchange representation of REMO is inadequate. Of the references given (p3, line 22), Kotlarski (2007) is a PhD thesis and Rechid (2009) is no longer available at the URL cited. In any case some description of the main points should be given here rather than relying solely on references.

We have removed the citation to the old link and improved the description part. Revised manuscript: Sec 2.1

**GMDD**
The description of tile structure is also confusing. It is stated that there are 3 tiles: land, ice and water (p3, line 23), but then that the standard land-surface scheme does not have a tile for lakes (p3, line 25). So does "water" mean "sea water", as on p4, line 15?

The original water tile included all water, including sea, lakes and rives. We changed this part to "REMO's surface pre-processor, which creates the surface related parameters including the fractions of different tiles (land, water and ice), was modified in this work to include also the lake fraction as an output variable. Thus, while the standard version of REMO lumps sea, lakes and rivers together in the water tile, here a separate tile is created for lakes and rivers." Revised manuscript: P4L23-27

**Section 2.3**

**P4, line 19: The GLCC lake fractions were used instead of those from Choulga et al (2014), although the latter are consistent with the depths taken from the same dataset. How do the two lake-fraction datasets compare?**

There are some small differences, but they don't make a big impact overall (the GLCC also sets Black Sea and Caspian Sea as lakes, so for larger European domains we use slightly modified GLCC data, where these areas are classified as seas). We could not use the Choulga et al. (2014) lake fractions, because this would have led to situations where the sum of tile fractions would have not been exactly one. Thus, the only way was to use the original GLCC fractions and use Choulga et al. (2014) for depths when available.

P4, lines 25–26: The fractional ice-cover in the original model is replaced with binary ice cover when FLake is implemented. It is possible that this change on its own may affect the model results. Its impact should have been estimated.

It is true that this will have some impact on the results, but we think impact is most

GMDD
likely small. First of all, the resolution of the driving data (here ERA-Interim, the resolution approximately 80 km) is coarser than the resolution used in the simulations. This means that when taking the sea-ice fraction from ERA-Interim, the values are basically either 0 or between 0.9 and 1. Thus, in practice, the ice fraction values are close to binary already in the standard version of the model.

Second, in reality, freezing is fairly fast during autumn and the biggest effect comes during spring, when the shores can be ice free and the rest of a lake is frozen (eventually ice breaks up and there is a mixture of open water and melting ice sheets). The albedo of water is smaller than the albedo of ice and therefore, the assumption of binary ice cover (i.e., ice fraction of 1 when there is any ice left) might delay the warming of lakes in spring in our simulations. However, the water temperature rise is still restricted by the ice in the lake (energy is used to melt the ice) so we don't consider this as a major issue. Also, in reality, there are some moisture fluxes from the open lake areas, but since the LWT is still quite low, these are also small. In terms of the manuscript, we have added more text and discussion about this issue. Revised manuscript: P5L7-14

P5, lines 8–15: There is some discussion of heat conductivity for lake ice and snow on lakes, but no discussion of how snow on land is modelled. Snow heat conductivity is taken as a single number (0.14 W/(mK), p5, line 14) despite the fact that snow density is apparently not fixed (p5, line 9). Studies such as Calonne et al. (GRL 2011) show that conductivity varies greatly with snow density, and the weighted sum of equation (1) presumably acts partly as a substitute model for this variation. It may be speculated that over-simplistic representation of snow conductivity generally may contribute to the winter cold bias even in the control run.

Eq. 1 is for snowy surfaces on ice. As described in the text, it is a mixture of snow and

**GMDD**
ice hear conductivity. The snow heat conductivity on land in the model depends on the snow temperature, so does the snow density. We have added more description of the model's surface scheme to section 2.1. Revised manuscript: P4L1-4

**Section 2.4**

Snow albedo gets a subsection here, although the rest of the snow scheme is not described. As stated later (e.g. p10, line 33), albedo changes may have little effect in winter given the lack of solar radiation in these regions then, so the emphasis on this aspect needs more justification.

The albedo was a natural step for further development as we had to add a snow albedo scheme for lakes anyways. As stated, these changes may have little effect, but it was one model development step that needed to be checked and document (similar albedo changes done for ECHAM6/JSBACH also motivated us). Unfortunately this did not fix the cold bias issue, but now we can rule it out from the list of the possible causes for the bias. We have improved this section in terms of justification. Revised manuscript: P6L11-16

**P5, line 29: The forest fraction mentioned here should have been described earlier in section 2.1.**

Yes, it could have been also discusses also in section 2.1. Section 2.1 describes REMO overall and here we wanted to show more details about the albedo calculations. Thus, REMO's snow albedo part is in section 2.4.

**P6, lines 3–11 seem to be a repeat of the previous paragraph.**

This paragraph has been removed.

**Section 2.5**

**GMDD**
**P6, line 32 – p7, line 2: The model end-state (in March?) is used as the model initial state (in August?). Would it not have been more suitable to use an initial state from the same season, say from August of the penultimate year? This part of the original text was misleading. Yes, we did match the months and we make it clearer in the text. Revised manuscript: P7L15-20**

P7, lines 5–6: Given that one comparison is with 2 m temperature, the height of the lowest model level should be stated, as also the surface interpolation scheme should earlier have been described.

The height varies from place to place, but is overall in 60 m height. We have also added missing information to section 3 explaining that we have done orographic correction to the model data (the temperatures have been corrected based on the height difference in each gridbox). We have also improved the description of the surface scheme. Revised manuscript: P3L29-31 and P7L21-23

**Section 2.6**

P7, line 31 - p8, line 1: Apparently 7 out of the 18 chosen study lakes may have unrepresentative mean depths in the model dataset. However, lake depth is often described as the parameter to which FLake is most sensitive. Is the study intended to test, or contradict, this assertion?

We use the default value of 7 meter when there is no data. This value, as explained in the text, is also from Choulga et al. (2014) and is representative for our domain. If we look the values in Table 2, where the mean depths are shown, we see that although some lakes do not have exact mean depth information, they do not differ much from the measured ones even with the default value. Indeed, some lakes have more a large error in their depth and this has been discussed as an error source when the LWT's are analyzed.

GMDD
**Section 2.6**

P 8, line 32. "The reason for the [control-run] cold bias is still unknown." Given that on p2 (lines 7–9) it is anticipated that a better lake representation will reduce wintertime heat fluxes, it seems odd to prioritise implementation of a lake model into a weather model with an existing, unexplained winter cold bias. Not least, it undermines the ice thickness/duration comparison. (As an aside, does the cold bias correlate with snow depth, as hinted by p9, lines 4–5?). The neglect of the cold-bias problem is acknowledged, but not really explained, in the last Conclusion paragraph. As a related point, the discussion on p11, lines 3–7 seems to say that the snow conductivity has been adjusted over lakes only, but acknowledges that snow thermodynamics may be a large factor in the cold bias. So why not try and improve "land" snow behaviour before applying FLake?

In any climate model, including REMO, there are many sources of error that could influence the simulated temperature, and often these errors partially compensate each other. For us, the treatment of lakes in REMO was an obvious part to start with, as the treatment of lake physics was known to be inadequate and to give rise to artificial heat and moisture sources. It is only after the lake physics was improved that the winter/spring cold bias in REMO became patently obvious (for the standard version, the underestimate was limited to the eastern part of the simulation domain, see the first two panels of Fig. 3). Thus our experiments demonstrate that the relatively good temperature simulation for standard REMO is a result of compensating errors, which is important to know.

This is not to say that the treatment of snow on land, including snow heat conductivity, should not be studied and improved. Indeed, our results for REMO-FL make this need more obvious. But, had we started with land snow physics, without improving the lakes, we would have had largely the same dilemma as we have now: the results could be compromised by the errors in the other parts of the model (in this case, the

**GMDD**
lake description). For example, hypothesizing that the cold bias is related to snow heat conductivity, tuning the land snow heat conductivity in a model with artificial heat sources from lakes could lead to misleading results. Overall, model development is done step by step, and including a lake model was one step that was needed for REMO.

We did check the correlation between snow depth and the cold bias, but no obvious correlation was found (in fact, they seemed to anticorrelate at many locations).

**Section 4.1**

P11, lines 27–32: "The underestimations in the model during spring can be mostly explained by these factors." The factors are discussed in a rather vague and unquantitative way, hence the proposed explanations are not very convincing.

We added more discussion in this paragraph and changed the ending. Revised manuscript: P12L28-P13L8

P12, lines 1–6: Errors are attributed to having the wrong mean lake depth, or to river input/output. It might have been better to adjust the depths of the study lakes, at least, to try and differentiate between these possibilities. Not least, it would be interesting to definitely attribute errors to river flow (or not), since this has a bearing on the general usefulness of 1-D models with no hydrological connections.

It is possible to change the depths, but we have not tried this in this work. Perhaps in a research with shorter simulations this would be a nice approach to see the model's sensitivity. The last point about the rivers is indeed interesting and should be investigated more. This would require higher spatial resolution and more detailed input information (or a module for rivers). GMDD
P12, lines 23–24: "Naturally, there are no thickness measurements for the beginning or end of the ice season as the ice is too shallow to make these manual measurements." Does this mean the thickness measurements are biased high? Yes, if there is a lot of variation on the seasons start and end times this can have a small effect to the near freezing and melting periods of the lakes. We will improve the text. Revised manuscript: P13L33-34

**P12, line 30 - p13, line 18; Again, the reasons for discrepancies are discussed in a qualitative way, with no quantitative testing or discrimination between the possible causes of error advanced.**

With our current setup the reasons discussed cannot be the separately tested. This is why we brought extra information from the snow depths over land nearby the lakes (supplementary). Also, the processes affecting snow depth vary from year to year, which can be seen from the variation of the measurements.

**Section 4.2**

**Figure 5: It appears that the surface temperature at Võrtsjärv is above freezing during some or all of the ice-covered period?**

Lake Võrtsjärv is located so south that it has a lot of year-to-year variation in ice cover. The measured ice period in Fig. 5 shows the average time when lake Võrtsjärv can be frozen. There are also years when the lake is open throughout or part of the winter, which explains why the average LWTs are above the freezing point.

**P13, line 28: "the main problem is the cold bias [of] the model." I believe this supports my earlier point about priorities.**

We agree that improving the cold bias in the model is an important research topic. However, see our response regarding step-wise model development above.

**GMDD**
And now we know more about this since lakes are much more realistic in the model which brings us back to the point-by-point model development. The next important step would indeed be to look into the cold bias issue, using knowledge gained in this study.

**Section 4.3**

Figure 6: I do not find the different (arbitrarily chosen?) depth scales for model and measurement profiles to be a useful way of making this comparison. Instead, profiles with a common depth scale should be shown. Then the authors can discuss (or the reader can decide) whether FLake gets the correct surface behaviour despite, or because of, its internal lake-structure model.

This is the only way as the model uses mean depths and measurements are from the deepest point. We did test of using the same scales, but this approach did not really tell anything about the vertical profiles. FLake's vertical profile is based on assumed-shape representation for the whole lake depth. If it is matched with real measurement from the deepest point using the modelled mean depth, we are not comparing the same things and as said, the figure has no informative value. Our approach is not perfect either, but it gives information about the profiles and how well the model captures them.

**Technical points**

Title: Should it be "freshwater lake model FLake" Corrected as suggested.

**p1, line 10: "are in realistic." ?** Removed "in".

**GMDD**
**p3, line 8: use NWP as defined on p1.**

Corrected as suggested.

**p4, line 11: Remove "on"** Corrected as suggested.

**p5 line 1: Does "It" refer to FLake?** No, it refers to the sediment module. Corrected to "This module..."

**Equation (1): hs does not seem to be defined.**

It is the snow depth in m. This information is now in the manuscript.

**p5, line 19: tp**  $\rightarrow$  **to** Corrected as suggested.

**p10, line 27: tot he**  $\rightarrow$  **to the** Corrected as suggested.

**GMDD**